# The Interrelationship of Benefit Finding, Demoralization, and Stigma among Patients with Parkinson’s Disease and Their Caregivers

**DOI:** 10.3390/healthcare12090878

**Published:** 2024-04-23

**Authors:** Pei-Chien Chou, Yu Lee, Yung-Yee Chang, Chi-Fa Hung, Ying-Fa Chen, Tsu-Kung Lin, Fu-Yuan Shih, Wu-Fu Chen, Pao-Yen Lin, Mian-Yoon Chong, Liang-Jen Wang

**Affiliations:** 1Department of Psychiatry, Kaohsiung Chang Gung Memorial Hospital and Chang Gung University College of Medicine, Kaohsiung 83301, Taiwan; ac0619@cgmh.org.tw (P.-C.C.); lyu722@cgmh.org.tw (Y.L.); chifa@cgmh.org.tw (C.-F.H.); paoyenlin@gmail.com (P.-Y.L.); 2Department of Neurology, Kaohsiung Chang Gung Memorial Hospital and Chang Gung University College of Medicine, Kaohsiung 83301, Taiwan; kcgmhcyy@cgmh.org.tw (Y.-Y.C.); alphac@cgmh.org.tw (Y.-F.C.); tklin@cgmh.org.tw (T.-K.L.); 3Center for Parkinson’s Disease, Kaohsiung Chang Gung Memorial Hospital, Kaohsiung 83301, Taiwan; 8902055@cgmh.org.tw (F.-Y.S.); ma4949@cgmh.org.tw (W.-F.C.); 4Department of Neurosurgery, Kaohsiung Chang Gung Memorial Hospital, Chang Gung University College of Medicine, Kaohsiung 83301, Taiwan; 5Health Management International, Singapore 218108, Singapore; chongmy@hmi.com.sg; 6Regency Specialist Hospital, Johor 81750, Malaysia; 7Department of Child and Adolescent Psychiatry, Kaohsiung Chang Gung Memorial Hospital and Chang Gung University College of Medicine, Kaohsiung 83301, Taiwan

**Keywords:** Parkinson’s disease, depression, caregivers, anxiety, stigma

## Abstract

Parkinson’s disease (PD) is a debilitating neurodegenerative disease with a relentlessly progressive course of illness. This study aimed to assess the dyadic dynamics of benefit finding (BF), demoralization, and stigma on the depression severity of PD patients and their caregivers. This study used a cross-sectional design with purposive sampling. In total, 120 PD patients and 120 caregivers were recruited from the neurological ward or neurological outpatient clinic of a medical center in Taiwan from October 2021 to September 2022. PD patients and their caregivers were enrolled and assessed using the Mini International Neuropsychiatric Interview, the Benefit Finding scale, Demoralization Scale, Stigma Subscale of the Explanatory Model Interview Catalogue, and Taiwanese Depression Questionnaire. Among the 120 patients and 120 caregivers that successfully completed the study, 41.7% (N = 50) and 60% (N = 72) were female, respectively. The most common psychiatric diagnoses of both the PD patients (17.5%) and their caregivers (13.3%) were depressive disorders. Using structural equation modeling, we found that the stigma, BF, and demoralization of PD patients might contribute to their depression severity. Demoralization and stigma of PD patients’ caregivers might also contribute to the depression severity of PD patients. Caregivers’ BF and demoralization were significantly linked with their depression severity. PD patients’ BF degree and their caregivers’ BF degree had significant interactive effects. Both patients’ and their caregivers’ stigma levels had significant interactive effects. Clinicians should be aware of and manage these contributing factors between PD patients and their caregivers in order to prevent them from exacerbating each other’s depression.

## 1. Introduction

Parkinson’s disease (PD) is a debilitating neurodegenerative disease with a relentlessly progressive course of illness, with an increasing incidence (ranged 10 to 50/100,000 person-years) and prevalence (ranged 100 to 300/100,000 population) with age globally [1,2,3]. PD is characterized by tremors, bradykinesia, rigidity, and postural instability [1]. Nonmotor symptoms such as anxiety, apathy, cognitive dysfunction, and depression were also noted in PD patients, with depression being the most prevalent and distressing symptom [4]. The impacts of untreated depression in patients with PD include greater morbidity, poorer quality of life (QoL), and increased mortality rates [5].

PD incidence and prevalence in Taiwan also has shown a remarkably increasing trend [3]. In a systemic analysis, the global burden of PD was found to have more than doubled from 1990 to 2016 [6]. PD patients become increasingly more dependent on personal care, which may have a variety of negative psychological effects on not only themselves but also their caregivers [7]. Caregivers of PD patients encounter various challenges to their physical, mental, and social well-being, with prior studies indicating that caring for a person with PD is linked to a higher likelihood of experiencing psychological distress, anxiety, and depression [8]. Therefore, healthcare professionals should also pay attention to the caregivers of PD patients, particularly in relation to their emotional state. Previous research has shown that the prevalence of depression among caregivers of Parkinson’s disease patients varied between 11% and 35% [9,10].

Benefit finding (BF) can be conceptualized as finding positive life changes or benefits from negative experiences such as trauma or chronic illness [11]. A 2022 cross-sectional study including 166 esophageal cancer patients showed that caregiver burden was negatively correlated with the Benefit Finding Scale (BFS) and positively correlated with the Hospital Anxiety and Depression Scale (HADS), which implied that BF may be crucial in buffering against anxiety and depression [12]. A 2014 cross-sectional study with 25 PD patient/spouse dyads showed that perceiving benefits from the experience of personally having or living with a spouse with PD was related to greater marital quality for both the patients and their spouses [13]. However, this study did not examine how BF impacted the depression of either the PD patients or their caregivers. Further studies are needed to answer this question.

Demoralization is characterized by a sense of incompetence through loss of meaning or purpose and may commonly be experienced among patients with terminal illness, chronic medical illness, or substance dependence [14]. A 2022 study that included 95 PD patients suggested that depression and anxiety may mediate between perceived stress and demoralization [15]. A 2022 cross-sectional study that sampled 142 end-of-life cancer patients found that both end-of-life patients and family caregivers may experience demoralization [16]. Demoralization is closely associated with depression in patients [17] and caregivers [16]; however, the above-mentioned studies did not reveal how demoralization impacted the depression of patients or their caregivers. A literature review found there is no dyadic study on the demoralization of PD patients and their caregivers. Advanced studies are needed to examine this important issue.

Stigma refers to the stereotypes or negative perceptions associated with a person or groups of people who are seen as different from or inferior to societal norms [18]. The results of a 2022 study on 196 PD patients suggested that depression or anxiety was associated with more perceived stigma and a poorer QoL [19]. A 2021 multinational study showed that physical symptoms such as bowel or bladder issues that accompany PD continued to be a cause of stigma for caregivers and families in the US [20]. However, the latter study from Henry did not show how stigma impacted the depression of PD patients’ caregivers. Further studies are warranted to elucidate this question.

Understanding how the BF, demoralization, and stigma of PD patients and their caregivers are related to their dyadic interaction may offer us more comprehensive perspectives on the treatment of PD patients and their caregivers’ depression and thereby reduce morbidity, increase QoL, and decrease the mortality rate. The global burden of PD may be lowered if we can come up with a more effective way to treat depression by understanding the interaction between patients and their caregivers.

There are very few studies focusing on BF, demoralization, or stigma of PD patients and their caregivers. Based on the above literature review, the hypothesis of this study was the existence of interactive effects regarding the depression of PD patients and their caregivers. The aims of our study were to examine the interaction between PD patients and their caregivers’ BF, demoralization, and stigma and the dyadic relationship of PD patients and their caregivers with these three factors and depression.

## 2. Methods

### 2.1. Participants 

This study was performed using a cross-sectional, purposive sampling design. Participants were recruited from the neurological ward or neurological outpatient clinic of a medical center from October 2021 to September 2022. Patient inclusion criteria included the following: (1) persons had to have been diagnosed with PD by an expert neurologist; (2) persons were able to realize the study procedure and could offer written informed consent. Patient exclusion criteria included: (1) persons with a diagnosis of delirium, or secondary parkinsonism, or atypical parkinsonism (e.g., dementia with Lewy bodies, progressive supranuclear palsy, etc.); (2) persons who were too weak to complete the questionnaire or clinical interview. 

Inclusion criteria for caregivers: (1) persons who were the principal caregivers for the patients. Principal caregivers were defined as “family members who are living with the patients and taking care of their daily needs”; (2) persons who were able to understand the study procedure and could offer written informed consent. Exclusion criteria for the caregivers: persons who were too weak to complete the questionnaire or clinical interview.

In total, 123 PD patients and their caregivers were invited to take part in this study initially. A total of 3 of the 123 PD patients were too weak to complete the questionnaire or clinical interview and three caregivers refused to participate in the interview. Data collection was completed for 120 PD patients and 120 caregivers, with a 97.7% response rate. A total of 89 (74.2%) of the 120 caregivers were spouses, 24 (20%) were children, and 7 (5.8%) were parents, siblings, or friends. 

### 2.2. Assessments

#### 2.2.1. Benefit Finding Scale (BFS)

The Benefit Finding Scale (BFS), developed by Tomich & Helgeson, is a tool specifically designed for breast cancer patients to evaluate how they perceive the positive impact that the experience of being diagnosed with and treated for breast cancer has had on their lives [21]. The BFS has been widely used with cancer patients and patients with other medical conditions, including PD [22]. The Chinese Benefit Finding Scale (CBFS) is a 6-dimensional, self-report inventory. Answers to each of the 22 items ranged from 1 (not at all) to 5 (extremely) on a 5-point Likert scale [23]. The CBFS has demonstrated good patient acceptability and exhibited satisfactory convergent validity, discriminant validity, concurrent validity, and internal consistency among Chinese patients with early-stage cancer [23]. 

#### 2.2.2. Demoralization Scale (DS)

The Demoralization Scale (DS), developed by Kissane et al., was used to detect the demoralization of cancer and other chronic patients. The DS identified 5 separate dimensions: loss of meaning, dysphoria, disheartenment, helplessness, and sense of failure [24]. These factors demonstrate satisfactory internal reliability and convergent validity. The DS contains 24 items that assess the frequency that participants have had specific feelings (0–4; never to all the time) during the past 2 weeks. Demoralization was defined as a DS score ≥24 [24]. The Mandarin version of the DS was validated by Hung et al., (2010), with satisfactory reliability and validity [25].

#### 2.2.3. Stigma Subscale of Explanatory Model Interview Catalogue (EMIC)

The EMIC is an anthropologically based interview instrument that examines patients’ patterns of distress, perceived cause, and help-seeking behavior to obtain quantitative and qualitative data [26]. The EMIC has been widely applied in the field of cultural psychiatry, which has focused on patients’ illness behavior and stigma during the last two decades [27,28]. Recently, the stigma subscale of the EMIC was used in a study on validating the Chinese version of the Shame and Stigma Scale (SSS) in patients with head and neck cancer and showed good agreement with this group of cancer patients [29].

#### 2.2.4. Unified Parkinson’s Disease Rating Scale (UPDRS) (Scores Range from 0 to 144)

The UPDRS is the primary universally accepted rating scale used in clinical research on PD and is utilized to track the progression of PD over time [30]. The UPDRS contains 5 segments: (1) mentation, behavior, and mood, (2) activities of daily living (ADLs), (3) motor sections, (4) complications of therapy, and (5) Part V: Hoehn and Yahr Scale [31]. Each answer to the scale is evaluated by a medical professional who specializes in PD.

#### 2.2.5. Numeric Pain Rating Scale (NPRS) 

The Numeric Pain Rating Scale (NPRS) is a single-dimensional tool used to assess the severity of pain in adults, including individuals experiencing cancer-related pain [32]. The 11-point numeric scale ranges from “0” (e.g., “no pain”) to “10” (e.g., “worst pain imaginable”). Patients with cancer have shown acceptable validity using this scale in pain evaluation [33]. A study was conducted to examine the reliability of the NPRS over time, and the results showed that it had good stability for worst pain (r = 0.93) and average pain (r = 0.78) over a 2-day period [32]. 

#### 2.2.6. Taiwanese Depression Questionnaire (TDQ)

The TDQ is a culture-sensitive self-rated instrument for detecting possible depressive disorder in Taiwan [34]. The TDQ includes 18 items related to appetite, insomnia, mood, somatic discomforts, interest, pessimism, etc. A 4-point Likert scale (ranging from 0 to 3) was used to examine participants’ frequency and severity of depression. The TDQ had satisfactory validity in a study among patients with cancer [35].

#### 2.2.7. Mini International Neuropsychiatric Interview (MINI)

The MINI is a concise, structured interview schedule and was used to make psychiatric diagnosis [36]. It has good reliability and can be adopted by nonphysicians. The validity and reliability of the MINI has been detected using the Structured Clinical Interview for DSM-III-R Patients (SCID-P), with satisfactory results [37]. The assessment time is about 15–20 min.

### 2.3. Procedures

Study procedures were as follows: (1) once a referral from the in-charge doctor at the neurological ward or neurological outpatient clinic was received, the research assistant would initiate contact with the PD patients and their caregivers. After explaining the study’s objectives and procedures to potential participants, only those who signed an informed consent form were included in the study. (2) Both the PD patients and their caregivers were given the BFS, DS, EMIC, TDQ, NPRS, and MINI assessments. In addition, the PD patients were given the UPDRS assessment. (3) The MINI was used by one staff psychiatrist (Dr. Y. Lee) to obtain a psychiatric diagnosis. (4) Our trained research assistant collected the patients’ and caregivers’ demographic and clinical data. 

### 2.4. Statistical Analyses

Descriptive statistics were analyzed using SPSS for Windows V. 16.0. Chi-square and Student’s *t* tests were used to examine the differences in demographic data and to subsequently assess the clinical features of participants with and without depressive disorder. To evaluate the impact of BF, demoralization, and stigma on patients’ and caregivers’ depression, we demonstrated actor and partner effects by using the actor–partner interdependence model (APIM) together with a dyads regression model [38]. We assessed the APIM using structural equation modeling (SEM), which was analyzed using SPSS Amos 24.0 [39].

## 3. Results

Initially, 123 PD patients and their caregivers were recruited for this study. However, three patients and three caregivers did not complete the study due to resistance or refusal, so data were collected from 120 patients and 120 caregivers. The response rate was 97.6%. Of the 120 PD patients who successfully completed the study, 58.3% (N = 70) were males. The average age of these PD patients was 66.8 ± 8.5 years. Their mean educational level was 11.3 ± 4.5 years, 88.3% were married, and 20.0% were currently employed. The average duration of the disease at the time of data collection was 8.4 ± 7.2 years (Appendix A). Of the 120 caregivers that successfully completed the study, 60.0% (N = 72) were females. The average age of the caregivers was 60.8 ± 12.7 years. Their mean educational level was 12.3 ± 4.1 years, 88.3% were married, and 37.5% were currently employed. The average duration of caring at the time of data collection was 8.4 ± 7.1 years (Appendix A). 

The results showed that 55.8% of PD patients and 50% of caregivers had one or more physical illnesses. In total, 36 percent of patients and 19.2% of caregivers had used hypnotics in the past (Appendix A). The average UPDRS of the PD patients was 36.8 ± 16.9, and their average H&Y staging was 2.2 (±0.7) (Appendix A).

Patients with PD were more often males (58.3% vs. 40.0%, x^2^ = 8.07, *p =* 0.005), elderly (66.8 ± 8.5 vs. 60.8 ± 12.7, t = 5.35, *p* < 0.001), with fewer years of education (11.3 ± 4.5 vs. 12.3 ± 4.1, t = −2.25, *p* < 0.05), more unemployment (80.0% vs. 62.5%, x^2^ = 8.97, *p* < 0.05), higher NPRS scores (3.4 (0–10) vs. 2.5 (0–10), t = 3.02, *p* < 0.05), higher TDQ scores (7.80 (0–35) vs. 4.86(0–24), t = 4.63, *p* < 0.001), lower BFS total scores (70.2 ± 13.9 vs. 73.0 ± 13.5, t = −2.00, *p* < 0.05), and higher EMIC total scores (3.6 (0–27) vs. 2.5 (0–17), t = 2.11, *p* < 0.05) than their caregivers (Table 1). This result suggested that PD patients were more frequently males, with lower educational level, more jobless status, higher pain severity, higher severity of depression, lower benefit finding level, and higher severity of stigma than their caregivers.

The most common psychiatric diagnoses of the PD patients were rapid eye movement (REM) sleep behavior disorder (23.3%), followed by depressive disorders (17.5%), insomnia disorder (10.8%), and anxiety disorders (8.3%). Among the depressive disorders, the most prevalent was other specified depressive disorder (11.7%), followed by major depressive disorder (MDD) (4.2%) and persistent depressive disorder (1.7%). Of the PD patients, 50.8% had a psychiatric diagnosis (Table 2 and Figure 1). 

The primary psychiatric diagnoses identified among the caregivers included depressive disorders (13.3%), anxiety disorders (12.5%), and insomnia disorders (3.3%). Within the category of depressive disorders, other specified depressive disorder (10.8%) was most prevalent, followed by MDD (2.5%). In all, 23.3% of the caregivers had a psychiatric diagnosis (Table 2 and Figure 1). 

Through the univariate analyses of the 120 PD patients, we were able to identify the factors that exhibited a significant association with depressive disorders. These factors included younger age (61.7 ± 8.6 vs. 67.8 ± 8.1, *p =* 0.002), younger age of onset (53.52 ± 11.01 vs. 59.32 ± 10.80, *p* < 0.05), anxiolytics/hypnotics use (13 (61.9) vs. 30 (30.3), *p* < 0.01), higher NPRS scores (4.7 ± 2.5 vs. 3.2 ± 2.7, *p* < 0.05), higher TDQ (17.0 ± 6.4 vs. 5.9 ± 4.4, *p* < 0.001), lower BFS total scores (63.5 ± 17.3 vs. 71.6 ± 12.7, *p* < 0.05), lower acceptance of BFS (9.1 ± 2.7 vs. 11.1 ± 2.1, *p* < 0.001), higher dysphoria of DS scores (7.3 ± 4.1 vs. 5.3 ± 2.0, *p* < 0.05), and higher EMIC total scores (8.9 (0–27) vs. 2.5(0–21), *p* < 0.01) (Appendix A). 

When analyzing the aforementioned significant factors in relation to depressive disorders using logistic regression analysis, higher anxiolytics/hypnotic use (OR = 3.28; 95% CI, 1.13–9.48; *p* = 0.028) and higher EMIC total scores (OR = 1.14; 95% CI, 1.06–1.23; *p* = 0.001) were two significant factors associated with depressive disorder in PD patients (Table 3). This result suggested that higher anxiolytics/hypnotic use and higher stigma severity were two associated factors of depression in PD patients. 

Through the univariate analyses of the 120 PD caregivers, we identified the following factors that exhibited a significant association with depressive disorders: anxiolytics/hypnotics use (6 (37.5) vs. 17 (16.3), *p =* 0.045), higher TDQ scores (15.8 (0–24) vs. 3.2 (0–17), *p* < 0.001), lower BFS total scores (64.5 ± 10.9 vs. 74.3 ± 13.5, *p* < 0.01), lower family relations BFS scores (6.1 ± 1.8 vs. 7.6 ± 1.6, *p =* 0.001), lower personal growth of BFS (20.5 ± 5.8 vs. 24.3 ± 5.3, *p* < 0.01), higher DS total scores (32.5 ± 14.2 vs. 24.4 ± 9.5, *p* < 0.05), higher dysphoria of DS scores (7.9 ± 4.1 vs. 5.1 ± 2.3, *p* < 0.05), and higher EMIC total scores (5.8 (0–15) vs. 2.0 (0–17), *p* < 0.05) (Appendix A).

When the above significant factors were analyzed relative to depressive disorders using logistic regression analysis, higher EMIC scores (OR = 1.16; 95% CI, 1.05–1.29; *p* = 0.006), lower BFS scores (OR = 0.96; 95% CI, 0.92–1.00; *p =* 0.039), and higher DS scores (OR = 1.07; 95% CI, 1.01–1.13; *p =* 0.021) were three significant factors associated with depressive disorder in PD caregivers (Table 4). This result demonstrated that higher stigma severity, lower BF level, and higher demoralization severity were three significant associated factors of depression among caregivers of PD patients.

Using SEM, we found that the stigma level (β = 0.19, *p* < 0.05), BF degree (β = −0.22, *p* < 0.01), and demoralization level (β = 0.27, *p* < 0.001) of PD patients were significantly linked with their depression severity (Figure 1). Furthermore, we found that caregivers’ stigma level (β = 0.19, *p* < 0.05) and demoralization level (β = 0.24, *p* < 0.001) were significantly linked with depression severity in PD patients. We also found that the BF degree (β = −0.19, *p* < 0.05) and demoralization level (β = 0.30, *p* < 0.001) of the caregivers were significantly linked with their depression severity (Figure 1). Furthermore, the PD patients’ BF degree and their caregivers’ BF degree had significant interactive effects (β = 0.34, *p* < 0.001), as did the patients’ stigma level and their caregivers’ stigma level (β = 0.43, *p* < 0.001) (Figure 2). 

## 4. Discussion

We found that REM sleep behavior disorder (RBD) (23.3%) was the most common psychiatric comorbidity in PD patients, followed by depressive disorder and insomnia. The prevalence of RBD in PD patients found in previous studies was inconsistent, varying significantly from 20% to 72% [40,41]. The inconsistent results in different studies may be due to the use of different diagnostic tools or inclusion criteria for RBD. In a Chinese study with 2462 PD patients in 2017, the overall prevalence of RBD symptoms in PD patients was 23.6%, consistent with the result in our study (23.3%). This similarity could be related to ethnic effect [42]. Of the caregivers of PD patients in our study, none had RBD, which indicated that the occurrence of RBD is more related to the clinical and pathophysiological relevance of RBD in PD patients. 

Depression is an important nonmotor symptom in PD, and its prevalence differs considerably, ranging from 20 to 50% among studies [43]. In our study, depressive disorder was the second most common psychiatric diagnosis among PD patients (17.5%) but the most common among caregivers (13.3%). The prevalence of depression among PD caregivers in our study was slightly lower than that in recent studies, which ranged from 14 to 35% [9]. The possible explanation for the lower depression morbidity in our study was the use of structured clinical interviews by psychiatrists, instead of self-rated depression questionnaires. Using a standardized clinical interview may lower the rate of false positive cases. Another possible explanation for the lower prevalence of depression in our study may be the pervasiveness of stoicism in Asian cultures [44]. The prevalence of depression in Taiwan, and also in other countries in the Asia-Pacific region, is generally lower than that in Western countries [45].

Our study found that the prevalence of depression in PD patients is greater than that in their caregivers. Patients with PD suffered from motor disability, the adverse effects of medications, and cognitive decline, which might worsen their depression [8]. Meanwhile, caregivers look after PD patients throughout the disease process; thus, having a high degree of depression morbidity might interfere with their caregiving ability and QoL [10]. In the present study, PD patients were more elderly, had a lower educational level, higher unemployment, higher pain severity, lower BF level, and a higher stigma severity than their caregivers. Caregivers of PD patients have established a high BF, and the positive life changes resulting from the struggle to cope with PD can help them overcome negative thinking from taking care of their patients [10,46]. The above-mentioned factors might provide a partial explanation for the higher depression morbidity of PD patients and is consistent with another finding of our study that the prevalence of depression in PD patients is higher than those among caregivers of PD patients [46]. 

In our study, the prevalence of depression is higher in PD patients than their caregiver. This may offer a possible explanation that PD patients may receive treatment such as antidepressants for their depression, which may mask their symptoms of anxiety disorder and lead to the lower percentage of anxiety in our study. We believe that more studies will be needed to clarify the prevalence of anxiety disorder in PD patients compared with that of their caregivers.

In our study, anxiety disorder was also a common psychiatric disorder in both PD patients and their caregivers at 8.3% and 12.5%, respectively. Anxiety severity is one of the associated factors of depression in PD patients and may also have been a possible risk factor for depression among the caregivers of PD patients in this study. Although the anxiety disorder morbidity of caregivers of PD patients was higher than that of PD patients, managing anxiety in both PD patients and caregivers is crucial to reducing their depression and improving their QoL. 

According to the literature review, most of the existing research on caregivers focuses on specific disorder or disease entities instead of general caregivers. For example, in a 2022 study about black and white hospice family caregivers, nearly one third of black and white caregivers reported anxiety symptoms, which showed a much higher prevalence of anxiety disorder in comparison to our study [47]. Possible explanations for the difference in our results and the above-mentioned study are differences in the research instrument, where we used standardized diagnostic interview that might lower the false positive rate. More studies are warranted to compare depression/anxiety of PD caregivers vs. other disease caregivers and general caregivers.

We found that lower BF level, higher demoralization severity, and higher stigma severity were associated with depression among PD caregivers. Several studies have shown a correlation between BF and depressive symptoms among caregivers [12,48]. In a 2013 UK study with 842 breast cancer patient caregivers, BF was significantly negatively correlated with the psychological distress of caregivers [48]. Another 2022 Chinese study of 228 esophageal cancer patient caregivers suggested that the caregivers’ BF played a mediating role in the anxiety–depression of the caregivers [12]. We suggested that a higher level of BF might lower the psychological distress of caregivers, which could lead to a lower prevalence of depression. Both of the afore-mentioned studies were in line with our result that BF was negatively correlated with the depression of the caregivers. This might be the first study to examine the effect of BF on depression in caregivers of PD patients. Further longitudinal studies are needed to elucidate the possible mediating effect of BF on depression. 

No current study has focused on the possible relationship of demoralization and depression in caregivers but, in the aforementioned 2022 study of 95 PD patients [15], depression may have been a mediator from perceived stress to demoralization. We hypothesized that this mediating effect also existed in caregivers. 

According to a 2018 study of 165 patients with MDD and their caregivers, caregiver depression may have been affected by the severity of patient depression and the personal stigma of the caregivers themselves [49]. This result partially supported our finding that stigma may be associated with the depressive disorder of caregivers. The stigma of a physical disorder such as PD or a mental disorder such as MDD may have an impact on caregiver depression. 

To investigate potential mechanisms connecting various factors and depression in patients with PD and their caregivers, we adopted SEM to examine APIM. With regard to actor effects, patients’ stigma, patients’ demoralization, and patients’ BF were significantly linked with the depression severity of PD patients; meanwhile, caregivers’ demoralization and caregivers’ BF were significantly linked to the depression severity experienced by caregivers. The following partner effects were found: (1) the caregivers’ demoralization (β = 0.24, *p* < 0.01) and stigma (β = 0.19, *p* < 0.05) were significantly linked with the depression severity of PD patients; (2) the patients’ stigma severity and their caregivers’ stigma severity. The patients’ BF level and their caregivers’ BF level had significant interactive effects. 

The PD caregivers’ stigma level was linked with the PD patients’ depression severity, but the PD caregivers’ stigma level was not linked with the caregivers’ depression severity. Based on this partner effect of caregivers’ stigma to patients’ depression severity, we suggest that caregivers’ stigma has a greater impact on patients’ depression than caregivers’ depression. Studies have shown that self-stigma may have a negative impact on the depression symptoms of patients with mental illness, which is consistent with our result showing that PD patients’ stigma is related to their depression severity [50,51]. We may hypothesize that PD caregivers tend to conceal their stigmatization, which may weaken the relationship between stigma and depression [52]. Moreover, caregivers’ demoralization level was linked with the depression severity of both the caregivers and their PD patients, and PD patients’ demoralization level was linked with their depression severity. This result suggested that demoralization had a solid impact and may contribute to depression severity.

The partner effect in our study not only existed in the link between caregivers’ stigma and demoralization and patients’ depression but also occurred between PD patients’ BF/stigma and their caregivers’ BF/stigma. According to a 2021 study of 286 pairs of colorectal cancer survivors and their caregivers, there was a positive correlation between dyads of patients’ and their caregivers’ BF [53]. This result was in line with our finding that PD patients’ BF and their caregivers’ BF had interactive effects. 

There was a positive relationship between PD patients and their caregivers’ stigma, and we suggested that this effect could be explained by the affiliate stigma in PD caregivers. According to the aforementioned 2021 study of PD symptoms and caregiver affiliate stigma, symptoms of PD related to bowel and bladder issues may be related to caregiver affiliate stigma [20]. We suggested that PD symptoms are not only related to patient stigma but also have an impact on caregiver stigma, and this may partially explain the positive relationship between PD patients’ and their caregivers’ stigma.

There are only a few dyadic studies that use APIM analysis with PD patients and their caregivers. Besides a study by Karlstedt et al., detecting the determining factor of a dyadic relationship and its psychological impact among PD patients and their spouses [54], our prior dyadic study on PD patients and their caregivers found that fatigue, resilience, and suicide ideation might contribute to the depression severity of both groups. Furthermore, the presence of suicidal ideation in patients had a negative impact on the level of depression experienced by their caregivers [48]. This is probably the first study to examine the effect of BF, demoralization, and stigma on depression severity of PD patients and their caregivers using an APIM.

There are several limitations to this study that should be mentioned. First, this was a cross-sectional study. The cause-and-effect relationship between BF, demoralization, and stigma and the depression of the patients and their caregivers could not be evaluated precisely, and further follow-up studies may be needed. Second, the patients were all from a general hospital and the sample size may not be large enough to represent the condition of the general population. Third, there are different tools to evaluate BF, demoralization, and stigma, which may cause variations when comparing our study with others. Furthermore, both BFS and DS were originally designed for patients with cancer. Whether these two scales can be applied to PD patients and their caregivers needs advanced studies to evaluate reliability and validity of BFS and DS. Fourth, some of the studies we referenced were published years ago. We may encounter bias due to the time gap between our study and others. Further studies with more recent databases are warranted to compare our findings. Fifth, our study design involved purposive sampling, which may have led to a sampling bias. However, the 97.6% response rate of the PD patients and caregivers compromised the effect of this limitation. Despite these limitations, our study is the first to detect the possible interrelationship of BF, demoralization, and stigma among patients with PD and their caregivers by using APIM. The use of a structured clinical interview by psychiatrists, instead of a self-reporting questionnaire, may have added objectivity to our study.

## 5. Conclusions

The clinical implications of this study are: (1) stigma, BF, and demoralization of PD patients might contribute to their depression severity. (2) Demoralization and stigma of PD patients’ caregivers might also contribute to the depression severity of PD patients. (3) Caregivers’ BF and demoralization were significantly linked with their depression severity. (4) PD patients’ BF/stigma degree and their caregivers’ BF/stigma degree had significant interactive effects. Clinicians need to recognize and address these factors that contribute to depression in both PD patients and their caregivers in order to prevent worsening of their depression. Further research is needed to determine if treating depression in both patients and caregivers is beneficial for both parties.

## Figures and Tables

**Figure 1 healthcare-12-00878-f001:**
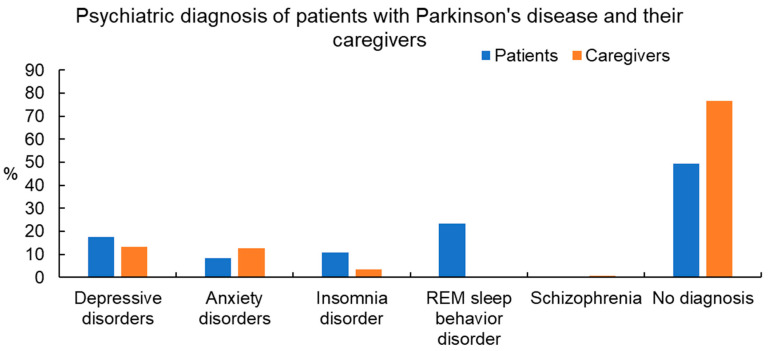
Psychiatric diagnosis of patients with Parkinson’s disease and their caregivers. REM sleep behavior disorder = rapid eye movement sleep behavior disorder. Data are presented as %.

**Figure 2 healthcare-12-00878-f002:**
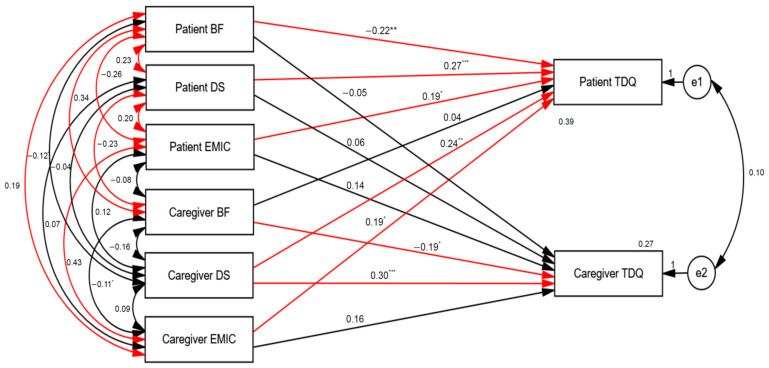
Structural equation modeling (SEM) of factors linked to depression in patients with PD and their caregivers. Model summary: chi-square = 0; df = 0; *p* = \*p*. The model fit: AGFI = \AGFI; RMSEA = \RMSEA; AIC = 72.00. * *p* < 0.05; ** *p* < 0.01; *** *p* < 0.001.

**Table 1 healthcare-12-00878-t001:** Demographic and clinical characteristics of the PD patients and their caregivers (N = 240).

	Patients N (%), N = 120	Caregivers N (%), N = 120	Total N (%), N = 240	Pair t/χ^2^	*p*
Gender				8.07	0.005
Male	70 (58.3)	48 (40.0)	118 (49.2)		
Female	50 (41.7)	72 (60.0)	122 (50.8)		
Age, years mean	66.76 ± 8.48	60.81 ± 12.73	63.78 ± 11.20	5.35	<0.001
Age of onset	58.29 ± 11.02				
Duration of PD (years)	8.43 ± 7.24				
Duration of caring (years)		8.40 ± 7.11			
Years of education	11.27 ± 4.47	12.32 ± 4.14	11.79 ± 4.33	−2.25	0.026
Education				3.44	0.064
Less than 12 years	41 (34.2)	28 (23.3)	69 (28.7)		
More than or equal to 12 years	79 (65.8)	92 (76.7)	171 (71.3)		
Marital Status				0	1.00
Unmarried	14 (11.7)	14 (11.7)	28 (11.7)		
Married	106 (88.3)	106 (88.3)	212 (88.3)		
Unemployed	96 (80.0)	75 (62.5)	171 (71.3)	8.97	0.003
Comorbid with other diseases	67 (55.8)	60 (50.0)	127 (52.9)	0.82	0.37
Suicide history	2 (1.7)	2 (1.7)	4 (1.7)	0	1.00
Anxiolytics/Hypnotics use	23 (19.2)	43 (35.8)	66 (27.5)	8.36	0.004
Family psychiatric history				0.34	0.56
No psychiatric history	118 (98.3)	119 (99.2)	237 (98.8)		
Depressive disorder	2 (1.7)	1 (0.8)	3 (1.3)		
Family suicide history	0	2 (1.7)	2 (0.8)	2.02	0.16
NPRS	3.44 (0–10)	2.49 (0–10)	2.97 (0–10)	3.02	0.003
UPDRS total scores	36.78 ± 16.91				
H&Y staging	2.23 ± 0.65				
TDQ	7.80 (0–35)	4.86 (0–24)	6.33 (0–35)	4.63	<0.001
BFS total scores	70.17 ± 13.87	73.03 ± 13.52	71.60 ± 13.74	−2.00	0.048
Acceptance	10.72 ± 2.31	10.96 ± 2.54	10.84 ± 2.42	−0.87	0.39
Family Relations	7.57 ± 1.89	7.40 ± 1.72	7.48 ± 1.81	0.85	0.40
World View	10.60 ± 3.36	11.60 ± 3.37	11.10 ± 3.40	−2.58	0.011
Personal Growth	21.90 ± 5.85	23.83 ± 5.51	22.86 ± 5.75	−2.84	0.005
Social Relations	8.90 ± 3.86	8.66 ± 3.47	8.78 ± 3.66	0.63	0.53
Health Behavior	10.48 ± 2.64	10.58 ± 3.01	10.53 ± 2.82	−0.29	0.78
DS total scores	26.83 ± 9.74	25.44 ± 10.52	26.13 ± 10.14	1.04	0.30
Loss of meaning	4.93 ± 1.98	4.68 ± 2.51	4.80 ± 2.26	0.78	0.44
Dysphoria	5.69 ± 2.56	5.49 ± 2.78	5.59 ± 2.67	0.62	0.54
Disheartenment	5.35 ± 2.49	5.48 ± 2.95	5.41 ± 2.72	−0.35	0.73
Helplessness	4.62 ± 2.58	4.06 ± 2.04	4.34 ± 2.34	1.92	0.06
Sense of failure	4.98 ± 1.79	4.60 ± 2.16	4.79 ± 1.99	1.54	0.13
EMIC total scores	3.60 (0–27)	2.49 (0–17)	3.05 (0–27)	2.11	0.037

Note: category data are presented as N (%), continuous variables are presented as score ± SD.; abbreviations: PD = Parkinson’s disease; NPRS = Numerical Pain Rating Scale; UPDRS = Unified Parkinson’s Disease Rating Scale; H&Y staging = Hoehn and Yahr staging; TDQ = Taiwanese Depression Questionnaire; BFS = Benefit Finding Scale; DS = Demoralization Scale; EMIC = Stigma Subscale of Explanatory Model Interview Catalogue.

**Table 2 healthcare-12-00878-t002:** Psychiatric diagnoses of patients with Parkinson’s disease and their caregivers.

MINI Diagnosis	Patients, N = 120	Caregivers, N = 120
Depressive disorders	21 (17.5)	16 (13.3)
Major depressive disorder	5 (4.2)	3 (2.5)
Other specified depressive disorder	14 (11.7)	13 (10.8)
Persistent depressive disorder	2 (1.7)	0
Anxiety disorders	10 (8.3)	15 (12.5)
Other specified anxiety disorder	6 (5.0)	11 (9.2)
Generalized anxiety disorder	3 (2.5)	3 (2.5)
Panic disorder	1 (0.8)	1 (0.8)
Insomnia disorder	13 (10.8)	4 (3.3)
REM sleep behavior disorder	28 (23.3)	0
Schizophrenia	0	1 (0.8)
No diagnosis	59 (49.2)	92 (76.7)

Note: data are presented as N (%); abbreviations: MINI = Mini International Neuropsychiatric Interview; REM sleep behavior disorder = rapid eye movement sleep behavior disorder.

**Table 3 healthcare-12-00878-t003:** Associated factors of depressive disorder in patients: logistic regression analysis.

Item	β	S.E.	Wald	Odds Ratio	95% C.I.	*p*
Anxiolytics/Hypnotics use	1.19	0.54	4.81	3.28	1.13–9.48	0.028
EMIC	0.13	0.04	11.77	1.14	1.06–1.23	0.001

Abbreviations: EMIC = Stigma Subscale of Explanatory Model Interview Catalogue. S.E. = standard error; 95% C.I. = 95% confidence interval.

**Table 4 healthcare-12-00878-t004:** Associated factors of depressive disorder in caregivers: logistic regression analysis.

Item	β	S.E.	Wald	Odds Ratio	95% C.I.	*p*
EMIC	0.15	0.05	7.67	1.16	1.05–1.29	0.006
BFS	−0.05	0.02	4.25	0.96	0.92–1.00	0.039
DS	0.06	0.03	5.29	1.07	1.01–1.13	0.021

Abbreviations: EMIC = Stigma Subscale of Explanatory Model Interview Catalogue; BFS = Benefit Finding Scale; DS = Demoralization Scale; S.E. = standard error; 95% C.I. = 95% confidence interval.

## Data Availability

The data of the current study are available from the corresponding author on reasonable request.

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
