# Peer review of "The Interrelationship of Benefit Finding, Demoralization, and Stigma among Patients with Parkinson’s Disease and Their Caregivers"

_healthcare, 2024, doi:10.3390/healthcare12090878_

Round 1
Reviewer 1 Report
Comments and Suggestions for Authors
Please find the attached comments.
Inconsistent font format was observed in the supplementary data.
---
Abstract:
line 21-23 specify the country of study
Introduction:
line 40-42 Authors are recommended to add some information about epidemiology, incidence, prevalence, mortality rate...etc related to PD
line 62, 65, 72 In the introduction section authors do no need to list the numeric values of previous studies to address the gap.
Methods:
The ethical approval section is missing?
was this study approved by the relevant IRB? Provide the No#
2.1 Participants:
the authors are recommended to add information about the sampling technique, sample size calculation...etc
is there any missing data? if yes how was that treated?
Results:
Authors are recommended to summarize the significant findings of this study instead of replicating the data in tables. This will help streamline the presentation for readers
Discussion:
line 292-294 Authors are recommended to compare these findings with the findings of previous similiar studies
line 316-317 Justify this statement

Author Response
Dear Reviewer,
Abstract:
line 21-23 specify the country of study
- Thank you for your suggestion. We added the country in line 23.
“In total, 120 PD patients and 120 caregivers were recruited from the neurological ward
or neurological outpatient clinic of a medical center in Taiwan from October 2021 to September 2022.”
Introduction:
line 40-42 Authors are recommended to add some information about epidemiology, incidence, prevalence, mortality rate...etc related to PD
- Thank you for your comment. We have reflected this comment by modifying the description about prevalence and incidence in the next paragraph (line 41-42 and 47).
“Parkinson’s disease (PD) is a debilitating neurodegenerative disease with a relentlessly progressive course of illness, with an increasing incidence (ranged 10 to 50/100,000 person-years) and prevalence (ranged 100 to 300/100,000 population) with age globally [1-3].”
line 62, 65, 72 In the introduction section authors do no need to list the numeric values of previous studies to address the gap.
- We agree with you and have simplified and retained part of the descriptions to demonstrate our review of current studies about the relationship between benefit findings/demoralization and PD.
Methods:
The ethical approval section is missing?
was this study approved by the relevant IRB? Provide the No#
- Thank you for your comment. Ethical approval was obtained from the Human Research Ethics Committee of Chang Gung Memorial Hospital (IRB No. 202100363B0; approval date: April 09, 2021) (line 471-475).
- “Institutional Review Board Statement: This study was conducted in accordance with the Declaration of Helsinki. Ethical approval was obtained from the Human Research Ethics Committee of Chang Gung Memorial Hospital (IRB No. 202100363B0; approval date: April 09, 2021). Confidentiality of the information was maintained, and the data were recorded anonymously throughout the study.”
2.1 Participants:
the authors are recommended to add information about the sampling technique, sample size
calculation...etc
- Thanks for your comment. We have mentioned our sampling in line 104-106. We performed this study using purposive sampling design, and the possible problem of sample size was mentioned in our limitations in line 430-432.
“Second, the patients were all from a general hospital, and the sample size may not be large enough to represent the condition of the general population.”
is there any missing data? if yes how was that treated?
- You have raised an important question. We have mentioned our missing data in line 119-121. Three out of 123 PD patients were not able to complete the study due to physical problem, and 3 out of 123 PD caregivers refused to participate in the interview. Their incomplete data were not included in the study. The 3 PD patients who cannot complete the study were treated and followed up at their neurological outpatient clinic. Totally, 120 PD patients and 120 PD caregivers were involved, with a 97.6% response rate.
Results:
Authors are recommended to summarize the significant findings of this study instead of replicating the data in tables. This will help streamline the presentation for readers
- Thank you for your valuable comments. We have added summary on the result section.
“This result suggested that PD patients were more males, lower educational level, more jobless status, higher pain severity, more severity of depression, lower benefit finding level, and higher severity of stigma than their caregivers.”
“This result suggested that more anxiolytics/hypnotic use and higher stigma severity were 2 associated factors of depression in PD patients.”
“This result demonstrated that higher stigma severity, lower BF level, and higher demoralization severity were 3 significant associated factors of depression among caregivers of PD patients.”
Discussion:
line 292-294 Authors are recommended to compare these findings with the findings of previous similar studies
- Thank you for your comment. We agree that it is an important point to compare our finding with other previous studies. Our finding was comparable to those study in China (“In a Chinese study with 2462 PD patients in 2017, the overall prevalence of RBD symptoms in PD patients was 23.6%, consistent with the result in our study (23.3%)”). “This similarity could be related to ethnic effect.” (line 310-311)
line 316-317 Justify this statement
- Thank you for your suggestion. This statement is also consistent with our finding below, and we have added further description in line 335-338.
“The above-mentioned factors might provide a partial explanation for the higher depression morbidity of PD patients, and is consistent with another finding of our study that the prevalence of depression in PD patients is higher than those among caregivers of PD patients [46].”
Reviewer 2 Report
Comments and Suggestions for Authors
This study represents an important contribution to the academic and scientific community. I am placing here some observations, which should be taken into account by the reviewers, prior to its acceptance for publication. Can the Benefit Finding Scale (BFS) and Demoralization Scale (DS) instruments be used for people with Parkinson's Disease? Are there any other limitations to the study? What is the impact of this study on the scientific and academic community? I appreciate your attention to this matter.
Author Response
Dear Reviewer,
Comment: This study represents an important contribution to the academic and scientific community. I am placing here some observations, which should be taken into account by the reviewers, prior to its acceptance for publication. Can the Benefit Finding Scale (BFS) and Demoralization Scale (DS) instruments be used for people with Parkinson's Disease? Are there any other limitations to the study? What is the impact of this study on the scientific and academic community? I appreciate your attention to this matter.
- Thank you for providing these insights. Both BFS and DS were originally designed for patients with cancer, and most studies utilizing these two scales focused on cancer-related issues. Besides cancer, both scales were applied to patients with other illnesses in other studies, such as Parkinson’s disease and buruli ulcer, which were both listed in our reference. There are different tools to examine BF and demoralization, and we have acknowledged this issue as one of our limitations in line 433-436.
Comment: “Furthermore, both BFS and DS were originally designed for patients with cancer. Whether these two scales can be applied to PD patients and their caregivers needs advanced studies to evaluate reliability and validity of BFS and DS.”
- While approaching PD patients with depression, caregivers' perspectives on the patient's illness might be overlooked. According to our study, demoralization and stigma of PD patients’ caregivers might also contribute to the depression severity of PD patients and themselves. Possible interactive effects were noted between PD patients and their caregivers’ BF/stigma degree. These findings provide additional perspectives for evaluating and treating depression in PD patients, especially the importance of taking a proactive approach to assessing or caring for the mental and physical well-being of caregiver. The involvement of caregivers in the treatment of depression in PD patients might be helpful.
Reviewer 3 Report
Comments and Suggestions for Authors
Thank you for giving the opportunity to review the manuscript “The Interrelationship of Benefit Finding, Demoralization, and Stigma Among Patients with Parkinson’s Disease and Their Caregivers” (healthcare-2931541). Authors conducted a study with a cross sectional design and aimed to assess the dyadic dynamics of benefit finding, demoralization, and stigma on the depression severity of PD patients and their caregivers.
The subject of the paper is interesting. However some parts of the study must be carefully revised.
Line 131/142 please add number, not author.
The most important results of this study should be summarized in a figure. This is important for the reader so that important information can be generated at a glance.
The chosen methodology is correct, from which the results are derived. The discussion could be improved somewhat by using more current studies and their content.
It is particularly important that the authors work out the implications. Why are findings important in practice?
Comments on the Quality of English LanguageModerate editing of English language required.
Author Response
Dear Reviewer,
Comment: Thank you for giving the opportunity to review the manuscript “The Interrelationship of Benefit Finding, Demoralization, and Stigma Among Patients with Parkinson’s Disease and Their Caregivers” (healthcare-2931541). Authors conducted a study with a cross sectional design and aimed to assess the dyadic dynamics of benefit finding, demoralization, and stigma on the depression severity of PD patients and their caregivers.
The subject of the paper is interesting. However some parts of the study must be carefully revised.
Line 131/142 please add number, not author.
- Thank you for the comment. We have corrected the reference into number accordingly.
The most important results of this study should be summarized in a figure. This is important for the reader so that important information can be generated at a glance.
- Thank you for your valuable comments! The most important information of our study has shown in Figure 1 and Figure 2. Figure 2 showed the structural equation modeling (SEM) of factors linked to depression in patients with PD and their caregivers. We have added Figure 1 ro illustrate the psychiatry diagnoses of PD patients and their caregivers. Figure 1. Psychiatric diagnosis of PD patients and their caregivers.
The chosen methodology is correct, from which the results are derived. The discussion could be improved somewhat by using more current studies and their content.
- Thank you for your suggestion. We attempted to search for the latest literature for comparison but only few studies were found. We have reflected this comment by adding this point to our limitations (line 436-439).
“Fourth, some of the studies we referenced were published years ago. We may encounter bias due to the time gap between our study and others. Further studies with more recent databases are warranted to compare our findings.”
It is particularly important that the authors work out the implications. Why are findings important in practice?
- Thank you for your important question. While approaching PD patients with depression, caregivers' perspectives on the patient's illness might be overlooked. According to our study, demoralization and stigma of PD patients’ caregivers might also contribute to the depression severity of PD patients and themselves. Possible interactive effects were noted between PD patients and their caregivers’ BF/stigma degree. These findings provide additional perspectives for evaluating and treating depression in PD patients, especially the importance of taking a proactive approach to assessing or caring for the mental and physical well-being of caregiver. The involvement of caregivers in the treatment of depression in PD patients might be helpful.
Comments on the Quality of English Language: Moderate editing of English language required.
We have performed English language edition on attached file.
